# MultiRC: Joint Learning for Time Series Anomaly Prediction and Detection with Multi-scale Reconstructive Contrast

## Abstract

Many methods have been proposed for unsupervised time series anomaly detection. Despite some progress, research on predicting future anomalies is still relatively scarce. Predicting anomalies is particularly challenging due to the diverse reaction time and the lack of labeled data. To address these challenges, we propose MultiRC to integrate reconstructive and contrastive learning for joint learning of anomaly prediction and detection, with multi-scale structure and adaptive dominant period mask to deal with the diverse reaction time. MultiRC also generates negative samples to provide essential training momentum for the anomaly prediction tasks and prevent model degradation. We evaluate seven benchmark datasets from different fields. For both anomaly prediction and detection tasks, MultiRC outperforms existing state-of-the-art methods. The code is available at https://anonymous.4open.science/status/MultiRC-CCE6.

## 1 Introduction

With the advancement of Internet of things (IoT), an increase number of sensors are utilized in industrial facilities to collect data in the form of continuous time series, which realizes the monitoring of system status (Li et al., 2021a). The anomaly detection technology (Li et al., 2021b; Wen et al., 2022; Chen et al., 2021a) has been widely used, which locates system malfunctions by identifying anomalies in historical data (Figure 1a). Effectively detecting anomalies helps pinpoint the sources of faults and prevent the spread of malfunctions. However, anomaly detection can only identify issues after they have occurred, which cannot meet the need for preventive maintenance timely.

As a new practical problem, anomaly prediction aims to predict whether an anomaly will occur in the future by capturing the fluctuations at the current time. In IoT, anomaly prediction can prevent full-scale failures. Previous works (Yin et al., 2022; You et al., 2024) assume and have observed that anomalies in production often do not occur suddenly but evolve gradually. Different kinds of fluctuations exist before the real anomalies, where data change from normal to abnormal. As shown in Figure 1b, the yellow period exhibits fluctuation patterns beginning distinct from the past, indicating the occurrence of possible future anomalies. The time interval of fluctuations is called reaction time, and such fluctuations are called precursor signals of future anomalies (Jhin et al., 2023). Following previous works (Yin et al., 2022), we predict future anomalies by identifying precursor signals, and emphasize that anomalies without precursor signals are unpredictable.

Due to the costs and rarity of anomalies, anomaly labels are difficult to obtain. To solve this, anomaly detection is often modeled with self-supervised (*reconstructive* (Zhou et al., 2024) or *contrastive* (Yang et al., 2023)) learning. Reconstruction methods expect normal points to be accurately reconstructed, while anomalies exhibit large reconstruction errors. Contrastive approaches promote similarity in the representations of positive sample pairs. Despite the effectiveness of existing works in anomaly detection, these methods cannot be directly and effectively extended to anomaly prediction, as shown in the following challenges.

**Challenge 1:** Different anomalies may occur rapidly or slowly, thus resulting in reaction time with varying lengths. Reconstruction methods focus on reconstructing individual data points which struggle to capture gradual reaction time (Li et al., 2022; Audibert et al., 2020; Zhang et al., 2022a). Contrastive learning methods compare time series segments based on fixed time lengths which struggle

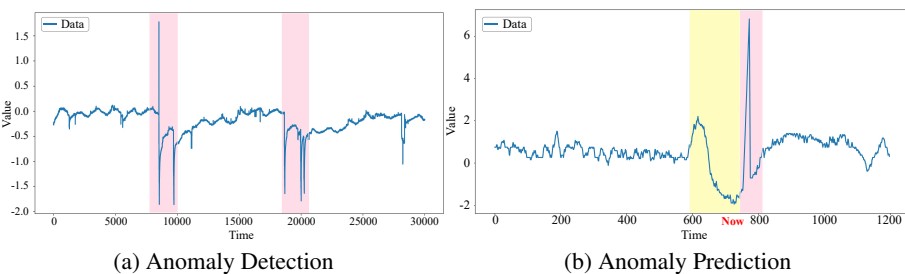

(a) Anomaly Detection  (b) Anomaly Prediction

Figure 1: (a) Anomaly detection on historical data. (b) Yellow period indicates the precursors where anomalies have not happened yet, and the pink period indicates future anomalies. Anomaly prediction forecasts if anomalies will occur in the future given current data.

to capture varying reaction time. Further, variates in the time series have different semantics, resulting in existence of distinct reaction time for different variates. Existing methods cannot identify fluctuations with such varying reaction time for different variates adaptively.

**Challenge 2:** Without labeled anomalies as negative samples, existing self-supervised methods will degrade into a trivial model (Ruff et al., 2018) that cannot learn meaningful information. For contrastive approaches, the absence of negative samples will lead to the learned features falling into a single mode where all features seem similar (Chen & He, 2021), thus failing to identify fluctuations. For reconstruction methods, a large model will degrade into identity transformation and a small model cannot learn complex temporal patterns (Wang et al., 2024), thus failing to assess the accurate magnitude of the fluctuations. Anomaly prediction requires not only identifying fluctuations but also assessing the magnitude of these fluctuations to predict the probability of future anomalies. Therefore, the existing methods fail to learn meaningful information for anomaly prediction.

In this work, we propose joint learning for time series anomaly prediction and detection with **multi-scale reconstructive contrast** (**MultiRC**), where a dual branch with joint reconstructive and contrastive learning is built upon a multi-scale structure. For **Challenge 1**, our novel multi-scale structure adaptively recognizes the varying reaction time for different variates with adaptive dominant period mask. Meanwhile, we exploit an asymmetric encoder-decoder to fuse cross-scale information. For **Challenge 2**, we incorporate controlled generative strategies to construct diverse precursors as negative samples to prevent model degradation, instead of only generating positive samples by data augmentation as in the existing contrastive methods (Woo et al., 2022; Zhang et al., 2022b). Specifically, our contrastive learning judges whether there are fluctuations by learning to distinguish positive and negative samples, while reconstruction learning assesses the magnitude of these fluctuations via learning to minimize the reconstruction errors for positive samples. Moreover, we provide a novel anomaly measurement for joint learning of anomaly prediction and detection, which helps anomaly prediction by detecting the degree of fluctuations in reaction time.

Our contributions are summarized as follows:

- A novel multi-scale structure is proposed to both reconstructive and contrastive learning, to adaptively recognize varying reaction time for different variates with adaptive dominant period mask.
- We propose controlled generative strategies to prevent model degradation and propose joint learning of anomaly prediction and detection with reconstructive contrast.
- For both anomaly prediction and anomaly detection tasks, MultiRC achieves state-of-the-art results across seven benchmark datasets.

## 2 RELATED WORK

**Multi-scale Learning.** Some multi-scale methods have been proposed for time series modeling (Chen et al., 2021b; Shen et al., 2020; Challu et al., 2022). THOC (Shen et al., 2020) learn multi-scale representations through different skip connections in RNN for each time point, but it cannot capture gradual change for continuous time intervals. DGHL (Challu et al., 2022) maps time series windows to hierarchical latent spaces, but it cannot capture cross-scale information from win-

dows of different lengths. Besides, none of these methods can adaptively recognize different scales or fuse multi-scale information for anomaly prediction.

**Time Series Anomaly Detection.** As a problem of practical significance, time series anomaly detection has remained a focal point in the fields of machine learning and data mining. Over recent years, researchers have used reconstruction paradigm (Su et al., 2019; Xu et al., 2021; Wang et al., 2024) or contrastive learning paradigm (Yang et al., 2023; Jhin et al., 2023) to delve into deeper data representations and patterns. In the reconstructive paradigm, OmniAnomaly (Su et al., 2019) captures the normal patterns of multivariate time series by learning their robust representations. Anomaly Transformer (Xu et al., 2021) proposes a minimax strategy that combines reconstruction loss to amplify the difference between normal and abnormal. $D^3R$ (Wang et al., 2024) combines decomposition and noise diffusion to directly reconstruct corrupted data. However, these studies either operate on a singular scale or perform indiscriminate reconstruction across the entire input range, limiting their flexibility and adaptability in extracting anomaly signals.

In the contrastive learning paradigm, DCdetector (Yang et al., 2023) introduces a dual-branch attention structure to learn a permutation invariant representation. However, it only focuses on relationships between positive samples, which greatly limits its performance in anomaly prediction tasks that require explicit anomaly labels. PAD (Jhin et al., 2023) directly uses resampling to generate artificial anomaly patterns during data preprocessing. This simple approach fails to adequately simulate the fluctuations of anomaly precursors. Although numerous contrastive paradigms (Yue et al., 2022; Lee et al., 2023; Zhang et al., 2022b; He et al., 2020) have been proposed in the field of time series analysis, they often merely treat the majority of time series segments as negative samples, which constrains the potential of contrastive learning to precisely identify anomalous patterns. We extract timing information at multiple scales to adapt to changes in reaction time. Furthermore, the model identifies fluctuations and amplitude of fluctuations through contrastive learning and reconstruction. Appendix B shows the architecture comparison of three approaches.

## 3 METHODOLOGY

**Problem Definition.** We denote the time series $\mathbf{X} \in \mathbb{R}^{T \times c}$ of length $T$, where each $\mathbf{x}_t \in \mathbb{R}^c$ is the observation at time $t$ and $c$ is the dimension of multivariate, such as the number of different sensors. The reaction time is a time interval $[t - r, t]$ with length $r$, where there are fluctuations in data $\mathbf{X}_{t-r:t}$ called the precursor, and there are real anomalies in future data $\mathbf{X}_{t+1:t+f}$. Data begin to change from normal to abnormal during the reaction time. The greater the degree of fluctuations, the higher the probability of future anomalies occurring. We emphasize that the anomalies without reaction time are unpredictable (Yin et al., 2022; Jhin et al., 2023).

For time series anomaly detection tasks, it takes the input $\mathbf{X}$ and outputs a vector $\mathbf{y} \in \mathbb{R}^T$ by sliding widows where $\mathbf{y}_t \in \{0, 1\}$ and 1 indicates an anomaly. For anomaly prediction, given a current time $t$ and the historical sub-sequence $\mathbf{X}_{t-h:t}$ of length $h$, the model outputs a probability score $\hat{\mathbf{p}}_t$, which indicates whether there is precursor in $\mathbf{X}_{t-h:t}$ or not and whether the future sub-sequence $\mathbf{X}_{t+1:t+f}$ will be anomalous, where $f \geq 1$ is the length of the look-forward window.

### 3.1 OVERALL FRAMEWORK

MultiRC consists of four modules (Figure 2). *Input sequence processing* normalizes the input multivariate time series through instance normalization and then uses channel-independence to split the input into univariate sequences. In Figure 2 we take one univariate sequence $\mathbf{x}$ as an example, and the other univariate sequences are modeled similarly. The *multi-scale structure* uses adaptive dominant period mask to adaptively recognizes varying reaction time for different variates, and then segments univariate sequences into patches of varying granularities. *Masked time series reconstruction* exploits asymmetric encoder-decoder to fuse multi-scale information and then evaluate the amplitude of fluctuations. *Generative-based contrastive learning* uses controlled strategies to construct negative samples to prevent model degradation. The encoder learns representations for distinguishing positive and negative samples, thus better judging the existence of fluctuation in the reaction time. Our encoder backbone consists of Transformer blocks (Vaswani et al., 2017) to extract temporal features from time series data.

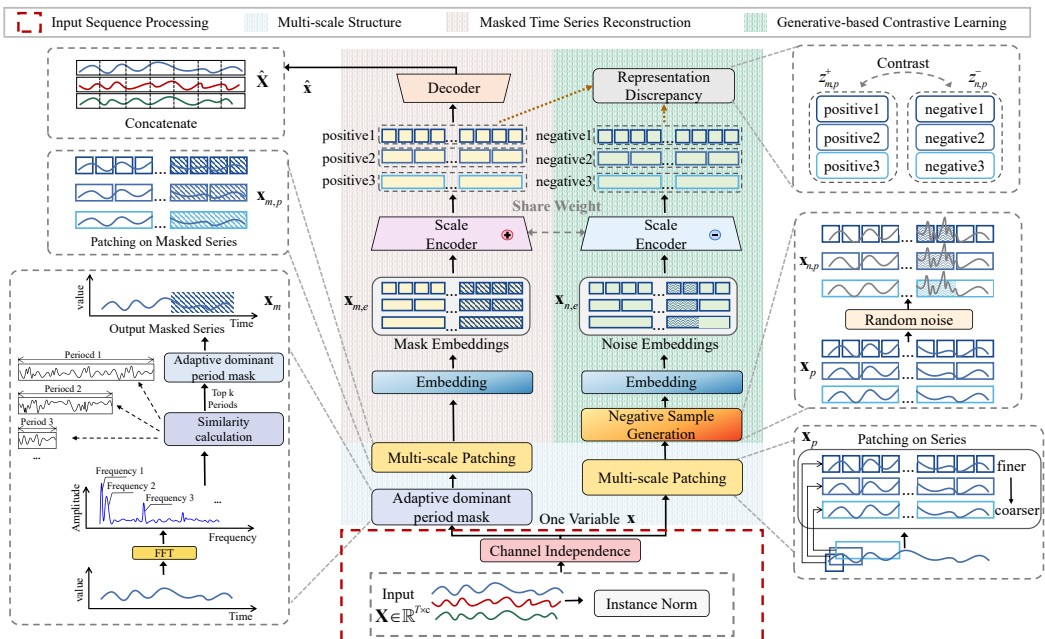

Figure 2: The overall architecture of MultiRC.

Overall, contrastive learning is used to judge the existence of fluctuation and reconstruction is used to assess the magnitude of the fluctuation, which both help indicate the probability of future anomalies. The anomaly score comprises the reconstruction error and the discrepancy between the representations of the positive samples.

## 3.2 MULTI-SCALE STRUCTURE

The assumption of the existence of reaction time (Yin et al., 2022; Jhin et al., 2023), where time series begin to change from normal to abnormal, ensures anomaly prediction. However, the duration of reaction time varies across different varieties and timestamps. To solve this, at each current time $t$, we explore the most dominant periods $\mathbf{q}_t$ for each univariate sequence in the frequency domain.

Specifically, the multi-scale structure contains *adaptive dominant period mask* and *multi-scale patching*. *Adaptive dominant period mask* takes each univariate sequence $\mathbf{x}$ as input and produces its masked sequence $\mathbf{x}_m$. Intuitively, one variate with longer periodic changes is more likely to evolve gradually while one variate with shorter periodic changes is more likely to evolve rapidly. Thus, we capture and mask with the most dominant periods for each variate to estimate the reaction time.

*Multi-scale patching* takes as input $\mathbf{x}$ and $\mathbf{x}_m$ in the dual branch, respectively. For *masked time series reconstruction*, $\mathbf{x}_m$ is fed into multi-scale patching; for *generative-based contrastive learning*, $\mathbf{x}$ is fed in. The multi-scale patching produces patched sequences $\{\mathbf{x}_{m,p}\}_{p=1}^a$ and $\{\mathbf{x}_p\}_{p=1}^a$ with different scales. This multi-scale patching is independent of the dominant period mask, which helps to simultaneously capture features from time intervals of different scales, thereby accommodating the varying reaction time that is inconsistent with the dominant periods.

**Adaptive Dominant Period Mask.** To capture the patterns of dominant periods, we first extract the periodic information in the frequency domain using Fast Fourier Transform (FFT) as follows:

$$\mathbf{A} = \mathrm{Amp}\left(\mathrm{FFT}(\mathbf{x})\right) \tag{1}$$

where $\mathrm{FFT}(\cdot)$ and $\mathrm{Amp}(\cdot)$ denote the FFT and the calculation of all amplitude values, and $\mathbf{A}$ represents the amplitude of each frequency. It is known in FFT that the frequency with larger amplitudes represents the more dominant period (Zhou et al., 2022). Thus, we calculate the frequency-based similarity between the historical univariate sub-sequences before the current time $t$ to select the most dominant periods for each variate. We use subscript $i$ and $j$ to represent different historical univariate sub-sequences before the current time $t$, e.g. $\mathbf{A}_i = \mathrm{Amp}\left(\mathrm{FFT}(\mathbf{x}_{t-h:t})\right)$ and

$\mathbf{A}_j = \text{Amp}\left(\text{FFT}(\mathbf{x}_{t-h-1:t-h-1})\right)$, and their frequency-based similarity is:

$$S_{i,j} = \frac{\mathbf{A}_i \cdot \mathbf{A}_j}{\|\mathbf{A}_i\|_2 \|\mathbf{A}_j\|_2} \tag{2}$$

Then we select the top-$k$ frequencies from the highest similarities:

$$i_{max}, j_{max} = \arg\max_{i,j}\left(S_{i,j}\right), \quad \mathbf{f} = \arg \text{Top-}k\left(\mathbf{A}_{i_{max}}, \mathbf{A}_{j_{max}}\right). \tag{3}$$

Specifically, $\mathbf{f} = \{f_1, \cdots, f_k\} \in \mathbb{R}^{k \times 1}$ is the top-$k$ dominant frequencies occurred most before the current time. The top-$k$ frequencies correspond to $k$ dominant period lengths $\mathbf{r} = \{r_1, \cdots, r_k\} \in \mathbb{R}^{k \times 1}$ where $r_k = \frac{1}{f_k}$. The dominant periods are used to estimate the reaction time.

The normal data can be reconstructed well and the fluctuations are hard to reconstruct (You et al., 2024; Campos et al., 2022). Thus, we focus on the reconstruction errors mainly during the reaction time to assess the magnitude of the fluctuations for anomaly prediction. We mask each univariate sequence $\mathbf{x}$ with adaptive mask length $r$, and $r$ is randomly sampled from $\mathbf{r}$:

$$\mathbf{x}_m = \mathbf{M} \odot \mathbf{x} \tag{4}$$

where $\|\mathbf{M}\|_1 = r$ and $\mathbf{M} = \{0, 0, \cdots, 1, \cdots, 1\} \in [0, 1]^T$ denotes the mask matrix near the current time with adaptive mask length, and $\odot$ indicates element-wise multiplication. Thus, we obtain the masked time series sequence $\mathbf{x}_m$ where each variate has varying mask lengths to estimate the reaction time.

**Multi-scale Patching.** Take the masked sequence $\mathbf{x}_m$ of the univariate sequence $\mathbf{x}$ as an example. We segment $\mathbf{x}_m$ into patches with multi-scale to enhance hierarchical information. Concretely, we obtain $a$ kinds of patch-based sequences from fine to coarse granularity upon $\mathbf{x}_m$, where $a$ represents the number of granularities. The multi-scale scaling process first generates a patch sequence $\mathbf{x}_1 \in \mathbb{R}^{N_1 \times P_1}$, where $P_1$ is the finest granularity patch size and $N_1$ is the number of patches. For the $i$-th scale, $1 < i \leq a$, we concatenate two adjacent patches from the $(i-1)$-th scale, and obtain sequence of the $i$-th scale containing $N_i$ patches with patch size $P_i$. By continuously grouping, we obtain $a$ sequences $\{\mathbf{x}_{m,p}\}_{p=1}^a$ with different scales. Similarly, we obtain $\{\mathbf{x}_p\}_{p=1}^a$ from the unmasked sequence, used for the contrastive branch.

### 3.3 MASKED TIME SERIES RECONSTRUCTION

To accurately assess the magnitude of fluctuations during the reaction time, we perform reconstruction on the dominant periods masked time series. This module is composed of asymmetric encoder-decoder architecture (as in the pink background section in Figure 2). The scale encoder is implemented based on the temporal transformer, and the output yields representations of positive sample pairs $\mathbf{z}_a^+ \in \mathbb{R}^{N_a \times d_{\text{model}}}$, where $d_{model}$ denotes the hidden state dimensions. The decoder reconstructs inputs of different scales using lightweight MLP to fuse multi-scale information, and outputs formatted as $\hat{\mathbf{x}} \in \mathbb{R}^{T \times 1}$.

**Scale Encoder.** There are $a$ encoder blocks, corresponding to patch sequences $\{\mathbf{x}_{m,p}\}_{p=1}^a$ at different scales passed through. The scale encoder is implemented based on the Transformer block. First, the embedded representation $\mathbf{x}_{m,e}$ is obtained via the embedding layer from each $\mathbf{x}_{m,p}$. Then, $\mathbf{x}_{m,e}$ is fed into multi-head self-attention layers, and the representations of different patches with different scales are units to learn temporal features across different time intervals. The scale encoder provides features $\{\mathbf{z}_p^+\}_{p=1}^a$ for both reconstruction and contrastive learning.

**Reconstruction Learning.** We concatenate the features $\{\mathbf{z}_p^+\}_{p=1}^a$ and use the MLP-based decoder to learn cross-scale information for reconstruction. The reconstruction result can be obtained by:

$$\hat{\mathbf{x}} = \text{Decoder}(\{\mathbf{z}_p^+\}_{p=1}^a) \tag{5}$$

MultiRC minimizes the mean squared error (MSE) loss. The losses across all channels are averaged to get the overall reconstruction loss $\mathcal{L}_{\text{Rec}}$:

$$\mathcal{L}_{\text{Rec}} = \frac{1}{c} \sum_{i=1}^c \|\mathbf{x} - \hat{\mathbf{x}}\|_2^2 \tag{6}$$

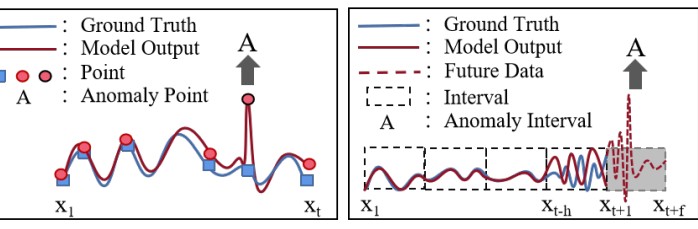

(a) Anomaly detection loss function    (b) Anomaly prediction loss function

Figure 3: Design specific loss functions for different tasks. $\mathbf{x}_{1:t}$ represents the historical time series, $\mathbf{x}_{t-h:t}$ represents the reaction time, and $\mathbf{x}_{t+1:t+f}$ denotes the size of the look-forward time window. In anomaly detection, significant loss differences indicate anomalies. In anomaly prediction, large fluctuations in loss during the reaction time suggest a high likelihood of future anomalies.

### 3.4 GENERATIVE-BASED CONTRASTIVE LEARNING

We introduce controlled generative strategies to construct diverse precursors as negative samples to prevent model degradation and better distinguish fluctuations. Multi-scale views from the masked time series reconstruction module are used as positive samples, while hard negative samples are generated through controlled noise pollution. The generative design of negative samples targets the need to avoid degradation and better judge fluctuations.

**Negative sample generation.** For multi-scale patch sequences $\mathbf{x}_p$, we apply different noise pollution strategies to generate negative samples $\mathbf{x}_{n,p}$. Different types of fluctuations are generated and the details are shown in Appendix D. The embedded representation $\mathbf{x}_{n,e}$ is obtained via the embedding layer from the generated negative samples $\mathbf{x}_{n,p}$. The encoder backbone is shared with the masked time series reconstruction module. Specifically, the generated data is fed into the backbone network to derive the hard negative sample representations $\mathbf{z}_p^- \in \mathbb{R}^{N_p \times d_{\mathrm{model}}}$ where $1 \leq p \leq a$.

**Representation discrepancy.** In anomaly prediction, we design an interval-wise contrastive loss function (Figure 3b). Three views are used as an example. Apply mean pooling to the various views generated by the scale encoder to obtain interval-level representations for each channel. Use the interval representations from the same channel $i$ as positive pairs $\mathbf{z}_{1[i]}^{pre}$, $\mathbf{z}_{2[i]}^{pre}$, $\mathbf{z}_{3[i]}^{pre}$. Similarly, use three views as an example. The hard negative samples $\mathbf{z}_{neg1}^{pre}$, $\mathbf{z}_{neg2}^{pre}$, $\mathbf{z}_{neg3}^{pre}$ and the representations from other channels are taken as negative pairs. The loss calculation $\mathcal{L}_{\mathrm{con}}^{inter}$ defined as:

$$\mathcal{L}_{\mathrm{con}}^{inter}(\mathbf{z}_1^{pre}) = -\frac{1}{c}\sum_{i=1}^{c} log \left( \frac{\exp(\mathbf{z}_{1[i]}^{pre} \cdot \mathbf{z}_{2[i]}^{pre}) + \exp(\mathbf{z}_{1[i]}^{pre} \cdot \mathbf{z}_{3[i]}^{pre})}{\sum\limits_{i \neq j}\sum\limits_{k=1}^{3} \exp(\mathbf{z}_{1[i]}^{pre} \cdot \mathbf{z}_{k[j]}^{pre}) + \sum\limits_{i} \exp(\mathbf{z}_{1[i]}^{pre} \cdot \mathbf{z}_{neg1}^{pre})} \right) \tag{7}$$

where $\exp(\mathbf{z}_{1[i]}^{pre} \cdot \mathbf{z}_{2[i]}^{pre})$ represents the inner product of the interval representations of $\mathbf{z}_1^{pre}$ and $\mathbf{z}_2^{pre}$. The overall contrastive objectives defined as follows, $H$ is the total number of different views:

$$\mathcal{L}_{Con} = \frac{1}{H}\sum_{i=1}^{H} \mathcal{L}_{\mathrm{con}}^{inter}(\mathbf{z}_i^{pre}) \tag{8}$$

For the anomaly detection task, we design a point-wise contrastive loss function (Figure 3a). Before calculating the loss, the outputs of encoders remain the dependence between each time point through additional upsampling:

$$\mathbf{z}^{det} = Upsampling(\mathbf{z}_{\xi(i,l_{\mathrm{orig}},l_{\mathrm{new}})}), \quad \xi(i, l_{\mathrm{orig}}, l_{\mathrm{new}}) = \left\lfloor \frac{i \times l_{\mathrm{orig}}}{l_{\mathrm{new}}} \right\rfloor \tag{9}$$

where $i \in \{0, 1, \cdots, l_{\mathrm{new}} - 1\}$ is the time index after upsampling. $\lfloor . \rfloor$ is the floor function. $z$ stands for $\mathbf{z}_p^+$ or $\mathbf{z}_p^-$. $l_{\mathrm{orig}}$ is the original length, that is, the number of time points in each patch. $l_{\mathrm{new}}$ is the target length, which is the size of the window size.

Representations of different views at the same time point $i$ are treated as positive pairs $\mathbf{z}^{det}_{1[i]}$, $\mathbf{z}^{det}_{2[i]}$, $\mathbf{z}^{det}_{3[i]}$. Negative pairs include: representations between different time points within the same view, the representation of any time point in $\mathbf{z}^{det}_1$ compared with all time points in $\mathbf{z}^{det}_2$ and $\mathbf{z}^{det}_3$ except that point, and the representation of any time point in the hard negative sample $\mathbf{z}^{det}_{neg1}$, $\mathbf{z}^{det}_{neg2}$, $\mathbf{z}^{det}_{neg3}$. The loss calculation $\mathcal{L}^{point}_{con}$ defined as:

$$\mathcal{L}^{point}_{con}(\mathbf{z}^{det}_1) = -\frac{1}{T}\sum_{i=1}^{T} log\left(\frac{\exp(\mathbf{z}^{det}_{1[i]} \cdot \mathbf{z}^{det}_{2[i]}) + \exp(\mathbf{z}^{det}_{1[i]} \cdot \mathbf{z}^{det}_{3[i]})}{\sum_{u \neq i}\sum_{k=1}^{3}\exp(\mathbf{z}^{det}_{1[i]} \cdot \mathbf{z}^{det}_{k[u]}) + \sum_{i}\exp(\mathbf{z}^{det}_{1[i]} \cdot \mathbf{z}^{det}_{neg1})}\right) \quad (10)$$

In summary, the various designs of the contrastive loss enable the same fundamental algorithm to be flexibly applied to different task requirements, enhancing the practicality of the model.

## 3.5 JOINT OPTIMIZATION

As previously mentioned, reconstruction and contrastive learning are interconnected. Reconstruction loss focuses on extracting key features from locally masked time series data. Contrastive loss effectively learn overall trends and patterns over time intervals. Let $\lambda$ be the weights that balance loss terms, the overall loss function is given:

$$\mathcal{L} = \lambda_{Con}\mathcal{L}_{Con} + \lambda_{Rec}\mathcal{L}_{Rec} \quad (11)$$

## 3.6 MODEL INFERENCE

During the inference phase, labeled negative sampled construction is not needed. The anomaly score is composed of the MSE between the input and the reconstructed output, as well as the representational distance between positive sample pairs, which can be determined using metrics such as Euclidean distance. The final anomaly score $f(x)$ is as follows:

$$f(x) = MSE(\hat{\mathbf{X}}, \mathbf{X}) + Dist(\mathbf{z}^+_p, \mathbf{z}^+_j)_{p \neq j, p, j=1,..,a} \quad (12)$$

which is a point-wise anomaly score. With a threshold (Wang et al., 2024), we can determine whether a point is abnormal. For anomaly prediction, the probability score $\hat{\mathbf{p}}_t$ is the averaged anomaly score from the look-back window. Apply a threshold (Yang et al., 2023), we can convert $\hat{\mathbf{p}}_t$ into a binary labels, if $\hat{\mathbf{p}}_t \geq \mu$, the future sub-sequence being anomaly.

# 4 EXPERIMENTS

## 4.1 EXPERIMENTAL SETTINGS

**Datasets.** We evaluated MultiRC on seven real-world datasets: (1) MSL (Mars Science Laboratory rover) (Hundman et al., 2018) includes operational data from multiple sensors on the Mars rover. (2) SMAP (Soil Moisture Active Passive satellite) (Hundman et al., 2018) provides soil moisture information collected from satellite sensors. (3) SMD (Server Machine Dataset) (Su et al., 2019) is a large-scale dataset collected over five weeks from a large Internet company. (4) PSM (Pooled Server Metrics) (Abdulaal et al., 2021) comprises data collected from eBay's application server nodes. (5) SWaT (Secure Water Treatment) (Li et al., 2019) is a dataset for security research in water treatment systems. (6) NIPS-TS-SWAN (Lai et al., 2021) is a comprehensive multivariate time series benchmark extracted from solar photospheric vector magnetograms. (7) NIPS-TS-GECCO (Rehbach et al., 2018) covers data collected from multiple sensors in a drinking water supply system. The training and validation data were split in an 8:2 ratio. Additional details on the datasets can be found in Appendix A.1.

**Baselines.** We compare our method with PAD (Jhin et al., 2023). We also modify unsupervised anomaly detection methods into anomaly prediction methods following PAD, including the reconstruction-based methods: CAE-Ensemble (Campos et al., 2022), GANomaly (Du et al., 2021),

Table 1: Anomaly prediction results for the five real-world datasets. Superior performance is indicated by higher metric values, with the top F1 scores emphasized in bold.

| Method | MSL | | | SMAP | | | SMD | | | PSM | | | SWaT | | | Average |
|--------|-----|-----|-----|------|-----|-----|-----|-----|-----|-----|-----|-----|------|-----|-----|---------|
| | P | R | F1 | P | R | F1 | P | R | F1 | P | R | F1 | P | R | F1 | F1 |
| DAGMM | 13.94 | 17.04 | 15.33 | 11.91 | 22.00 | 15.51 | 14.74 | 15.37 | 15.05 | 39.27 | 36.14 | 37.64 | 82.14 | 61.72 | 70.48 | 30.80 |
| Iforst | 13.82 | 17.11 | 15.29 | 11.79 | 18.67 | 14.45 | 15.25 | 18.28 | 16.63 | 39.39 | 38.82 | 39.10 | 75.39 | 62.22 | 68.17 | 30.72 |
| Deep SVDD | 12.94 | 16.04 | 14.32 | 11.77 | 17.09 | 13.94 | 14.99 | 15.78 | 15.37 | 36.63 | 36.30 | 36.46 | 81.29 | 61.95 | 70.31 | 30.08 |
| A.T. | 14.62 | 12.42 | 13.43 | 11.40 | 22.80 | 15.20 | 10.52 | 19.42 | 13.65 | 30.56 | 34.64 | 32.47 | 75.00 | 60.50 | 66.97 | 28.34 |
| DCdetector | 2.35 | 3.46 | 2.80 | 8.04 | 11.30 | 9.40 | 3.71 | 9.03 | 5.26 | 28.97 | 12.05 | 17.02 | 46.45 | 36.36 | 40.79 | 15.05 |
| PAD | 14.37 | 13.22 | 13.77 | 15.19 | 34.41 | 21.08 | 11.23 | 18.76 | 14.05 | 31.23 | 33.05 | 32.11 | 74.83 | 59.09 | 55.04 | 27.21 |
| Omni | 13.66 | 21.66 | 16.75 | 11.90 | 23.39 | 15.77 | 16.22 | 18.19 | 17.15 | 39.45 | 40.88 | 40.15 | 85.73 | 58.57 | 69.59 | 31.88 |
| GANomaly | 15.36 | 17.30 | 16.27 | 12.18 | 23.41 | 16.02 | 15.06 | 21.39 | 17.68 | 37.02 | 43.06 | 39.81 | 83.99 | 59.77 | 69.84 | 31.92 |
| CAE-Ensemble | 20.35 | 18.27 | 19.26 | 15.88 | 26.68 | 19.91 | 18.41 | 19.51 | 18.94 | 40.18 | 41.99 | 41.07 | 88.61 | 59.90 | 71.48 | 34.13 |
| D³R | 19.99 | 20.32 | 20.15 | 15.73 | 26.89 | 19.85 | 15.78 | 19.33 | 17.38 | 40.73 | 42.21 | 41.46 | 84.13 | 61.85 | 71.29 | 34.02 |
| MultiRC | 14.74 | 69.84 | **24.34** | 16.59 | 47.11 | **24.53** | 19.99 | 23.27 | **21.51** | 36.77 | 65.30 | **47.05** | 99.38 | 60.29 | **75.05** | **38.49** |

OmniAnomaly (Su et al., 2019), Anomaly Transformer (A.T.) (Xu et al., 2021), D³R (Wang et al., 2024); the contrastive learning methods: DCdetector (Yang et al., 2023), PAD (Jhin et al., 2023); the classic methods: IForest (Liu et al., 2008); the density-estimation method: DAGMM (Zong et al., 2018); the clustering-based method: Deep SVDD (Ruff et al., 2018). We have used all baselines with their official or open-source versions. Additional details on the baselines can be found in Appendix A.2.

**Evaluation criteria.** We use the metrics Precision(P), Recall(R), and F1-score(F1) for comprehensive comparison (Su et al., 2019). In anomaly prediction tasks, we predict future window anomalies (Jhin et al., 2023), for which we use classic metrics. With respect to anomaly detection tasks, we detect point anomaly as previous works (Wang et al., 2024). As pointed out in (Kim et al., 2022), the point adjustment strategy commonly used in previous works (Song et al., 2024; Yang et al., 2023; Zhao et al., 2020; Xu et al., 2021) is unreasonable. This strategy assumes that if one anomaly is correctly detected within a continuous anomalous segment, then all points in that segment are considered correctly detected. The recently proposed affiliation-based P/R/F1 (Huet et al., 2022; Wang et al., 2024) provides a reasonable evaluation for anomaly detection in time series, so we employ this for comparison.

## 4.2 MAIN RESULTS

Tables 1 and Table 2 respectively present the performance comparison of the model in anomaly prediction and detection tasks. As indicated by the tables, on all five real-world datasets, MultiRC surpasses the adversary algorithms, achieving the best F1 and Aff-F1 performance, thereby confirming its effectiveness. Performance comparisons on the NIPS-TS-SWAN and NIPS-TS-GECCO datasets are presented in Appendices F and G. The visualization results of anomaly prediction are presented in Appendix E.

**Anomaly prediction.** MultiRC has achieved improvements of 4.19% (from 20.15 to 24.34), 4.68% (from 19.85 to 24.53), 2.57% (from 18.94 to 21.51), 5.59% (from 41.46 to 47.05), and 3.57% (from 71.48 to 75.05) on the MSL, SMAP, SMD, PSM and SWaT datasets, respectively.

Traditional machine learning methods, such as IForest, often underperform in generalization because they do not take into account the continuity and complexity of time series data. Despite being based on contrastive learning, the performance of DCdetector is not satisfactory. This is because it focuses only on positive samples, resulting in severely limited performance in anomaly prediction tasks that require label support from negative samples. Compared to the previous state-of-the-art model, D³R, MultiRC significantly improves the F1 scores on benchmark datasets. In our model, multi-scale mask reconstruction and multi-noise contrastive paradigm are combined. MultiRC takes into account the periodic variations in time series and further enhances hierarchical information through a multi-scale structure, which facilitates the identification of different reaction time. The encoder outputs are simultaneously used for both reconstruction and contrastive tasks, making the learned feature representations more meaningful. These designs address the shortcomings of previous work and maintain outstanding performance across different datasets. Furthermore, the performance on the PSM and SWaT datasets is significantly better than on other datasets, likely because anomaly prediction requires the model to detect precursor signals. In some datasets, these precursors are less apparent, thereby limiting the model performance.

Table 2: Experimental results for the anomaly detection on five time-series datasets. The best Aff-F1 scores are highlighted in bold.

| Method | MSL Aff-P | Aff-R | Aff-F1 | SMAP Aff-P | Aff-R | Aff-F1 | SMD Aff-P | Aff-R | Aff-F1 | PSM Aff-P | Aff-R | Aff-F1 | SWaT Aff-P | Aff-R | Aff-F1 | Average Aff-F1 |
|---|---|---|---|---|---|---|---|---|---|---|---|---|---|---|---|---|
| DAGMM | 4.07 | 92.11 | 68.14 | 50.75 | 96.38 | 66.49 | 63.57 | 70.83 | 67.00 | 68.22 | 70.50 | 69.34 | 59.42 | 92.36 | 72.32 | 68.65 |
| IForest | 53.87 | 94.58 | 68.65 | 41.12 | 68.91 | 51.51 | 71.94 | 94.27 | 81.61 | 69.66 | 88.79 | 78.07 | 53.03 | 99.95 | 69.30 | 69.82 |
| Deep SVDD | 49.88 | 98.87 | 65.73 | 42.67 | 68.23 | 51.94 | 65.84 | 80.43 | 72.58 | 58.32 | 60.11 | 59.95 | 55.73 | 97.34 | 70.77 | 64.19 |
| A.T. | 51.04 | 95.36 | 66.49 | 56.91 | 96.69 | 71.65 | 54.08 | 97.07 | 66.42 | 54.26 | 82.18 | 65.37 | 53.63 | 98.27 | 69.39 | 67.86 |
| DCdetector | 55.94 | 95.53 | 70.56 | 53.12 | 98.37 | 68.99 | 50.93 | 95.57 | 66.45 | 54.72 | 86.36 | 66.99 | 53.25 | 98.12 | 69.03 | 68.40 |
| PAD | 56.33 | 82.21 | 68.15 | 41.67 | 64.52 | 53.94 | 59.54 | 67.66 | 63.71 | 68.45 | 57.72 | 59.21 | 54.73 | 92.35 | 68.06 | 62.61 |
| Omni | 51.23 | 99.40 | 67.61 | 52.74 | 98.51 | 68.70 | 79.09 | 75.77 | 77.40 | 69.20 | 80.79 | 74.55 | 62.76 | 82.82 | 71.41 | 71.93 |
| GANomaly | 56.36 | 98.27 | 68.01 | 56.44 | 97.62 | 72.52 | 73.46 | 83.25 | 74.20 | 55.31 | 98.34 | 75.11 | 59.93 | 81.52 | 70.24 | 72.01 |
| CAE-Ensemble | 54.99 | 93.93 | 69.37 | 62.32 | 64.72 | 63.50 | 73.05 | 83.61 | 77.97 | 73.17 | 73.66 | 73.42 | 62.10 | 82.90 | 71.01 | 71.05 |
| $D^3R$ | 66.85 | 90.83 | 77.02 | 61.76 | 92.55 | 74.09 | 64.87 | 97.93 | 78.02 | 73.32 | 88.71 | 80.29 | 60.14 | 97.57 | 74.39 | 76.76 |
| MultiRC | 67.94 | 93.03 | **78.53** | 66.66 | 89.94 | **76.57** | 75.36 | 94.85 | **83.99** | 73.36 | 93.13 | **82.08** | 62.52 | 97.26 | **76.12** | **79.45** |

**Anomaly detection.** As can be seen from the Table 2, MultiRC also achieves optimal performance in the exception detection tasks. This is 1.51%-5.97% higher on average than previous SOTA methods. Additionally, many previous studies were evaluated through point adjustment, leading to a false boom. Metrics based on affiliation offer a more objective and reasonable assessment for various methods, resulting in lower scores for past approaches. Overall, MultiRC not only effectively detects anomalies within complex data but also assists in predicting future anomalies.

## 4.3 ABLATION STUDIES

In order to verify the effectiveness and necessity of our designs, we conduct ablation studies focusing on key components of our model design: the multi-scale structure, the masked time series reconstruction and the generative-based contrastive learning.

Table 3: Results of anomaly prediction ablation studies. The best scores are highlighted in bold.

| Dataset | MultiRC P | R | F1 | w/o multi-scale P | R | F1 | w/o adaptive mask P | R | F1 | w/o reconstruction P | R | F1 | w/o contrastive P | R | F1 | w/o generation P | R | F1 |
|---|---|---|---|---|---|---|---|---|---|---|---|---|---|---|---|---|---|---|
| MSL | 14.74 | 69.84 | **24.34** | 11.50 | 53.89 | 18.96 | 14.72 | 58.32 | 23.51 | 7.69 | 34.19 | 12.56 | 11.62 | 51.85 | 18.99 | 12.62 | 55.52 | 20.56 |
| SMAP | 16.59 | 47.11 | **24.53** | 15.48 | 31.76 | 20.82 | 14.94 | 39.95 | 21.75 | 14.94 | 26.92 | 19.22 | 16.63 | 40.41 | 23.56 | 16.37 | 40.05 | 23.24 |
| PSM | 36.77 | 65.30 | **47.05** | 42.39 | 50.27 | 46.00 | 35.84 | 62.00 | 45.42 | 28.54 | 35.60 | 31.68 | 36.33 | 65.17 | 46.65 | 37.03 | 63.79 | 46.86 |

Table 4: Ablation results of anomaly detection. The best scores are highlighted in bold.

| Dataset | MultiRC Aff-P | Aff-R | Aff-F1 | w/o multi-scale Aff-P | Aff-R | Aff-F1 | w/o adaptive mask Aff-P | Aff-R | Aff-F1 | w/o reconstruction Aff-P | Aff-R | Aff-F1 | w/o contrastive Aff-P | Aff-R | Aff-F1 | w/o generation Aff-P | Aff-R | Aff-F1 |
|---|---|---|---|---|---|---|---|---|---|---|---|---|---|---|---|---|---|---|
| MSL | 67.94 | 93.03 | **78.53** | 65.87 | 93.57 | 77.31 | 67.48 | 90.32 | 77.25 | 44.65 | 44.09 | 44.37 | 67.01 | 93.08 | 77.92 | 67.37 | 93.09 | 78.17 |
| SMAP | 66.66 | 89.94 | **76.57** | 64.28 | 90.34 | 75.11 | 66.41 | 88.31 | 75.81 | 53.74 | 76.61 | 63.17 | 65.39 | 90.07 | 75.77 | 65.65 | 86.16 | 74.52 |
| PSM | 73.36 | 93.13 | **82.08** | 76.15 | 80.59 | 78.31 | 72.29 | 90.38 | 80.33 | 52.94 | 69.90 | 60.25 | 73.22 | 92.43 | 81.71 | 73.53 | 92.74 | 82.03 |

In Table 3, we introduce our ablation experimental results for the anomaly prediction. We attempted to remove the multi-scale structure (w/o multi-scale), using only a single-scale sequence. Replace the adaptively adjusted mask scale with a fully random mask (w/o adaptive mask). We also remove the masked reconstruction module (w/o reconstruction), the contrastive learning module (w/o contrastive), and the noise generation variants (w/o generated samples).

The results show that performance decreased across all datasets after removing the multi-scale framework. Notably, some datasets experienced a significant drop in performance, which is due to the different levels of dependence on the multi-scale framework caused by varying response time lengths. The random mask led to a marked decrease in performance, indicating that adjusting the mask scale based on frequency and periodicity is beneficial for identifying different response times. Removing the masked reconstruction module or contrastive learning leads to the lack of ability to effectively identify fluctuations and magnitude, thus not maintaining the best performance on all datasets. The sample generation strategy has led to significant improvements (3.78%, from 20.56 to 24.34). This emphasizes the importance of hard negative samples in enhancing the ability to avoid degradation.

In the ablation studies for anomaly detection, we define the ablation model in exactly the same manner, results are shown in Table 4. This further substantiates the importance of each component to the overall efficacy of the model.

## 4.4 DISCUSSION AND ANALYSIS

**Analysis of reaction time.** For reaction time, we conducted an analysis of window size performance on three datasets, as shown in Figure 4. This study follows the methodologies mentioned in the literature (Schmidl et al., 2022; Wenig et al., 2022), adjusting the sliding window size. The results indicate that PSM achieves optimal performance within a small window size (16), suggesting a shorter reaction time and significant fluctuations within this dataset. In contrast, MSL and SMAP are evident over a longer time span (64), indicating a relatively long reaction time. This disparity underscores the necessity of adopting the Multi-scale structure to accommodate varying lengths of reaction time needs.

**Hyperparameter analysis.** The hyperparameters that might influence the performance of MultiRC include the hidden state dimension, batch size, and patch size. To analyze their impact on the results, we conducted a hyperparameter sensitivity analysis on the MSL, SMAP and PSM datasets. The findings are presented in Figure 5. This figure primarily depicts the results for anomaly prediction tasks. For the hyperparameter results of anomaly detection, see the Appendix G. Figure 5a demonstrates the performance across different sizes of the latent space, as the performance of many deep neural networks is affected by $d_{model}$. Figure 5b displays the outcomes

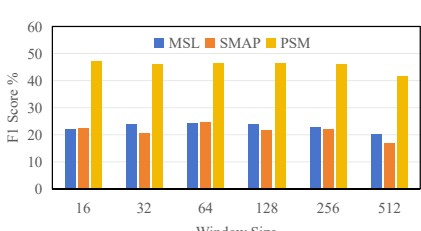

Figure 4: Performance of different datasets under different window sizes.

for MultiRC when trained with various batch sizes. Figure 5c displays the model performance at different patch sizes. The experimental results show that the performance of the model remains relatively stable in different patch size combinations. A more detailed comparison and analysis regarding multi-scale patch size can be found in Appendix C.

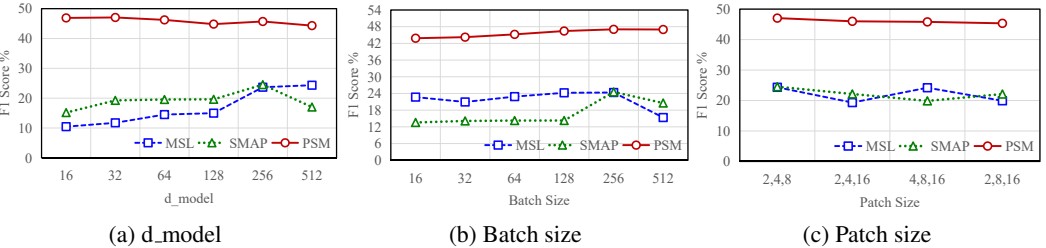

(a) d_model          (b) Batch size          (c) Patch size

Figure 5: Parameter sensitivity studies of main hyper-parameters in MultiRC.

## 5 CONCLUSION

We propose MultiRC, a multi-scale reconstructive contrast for time series anomaly prediction and detection tasks. Our multi-scale structure enables the model to adapt to precursor signals of varying reaction times. In addition, sample generation is used to construct hard negative samples to prevent model degradation, where our model learns more meaningful representations to better identify fluctuations with contrastive learning and evaluate the amplitude of fluctuations with reconstruction learning. Experimental results demonstrate that MultiRC surpasses existing works in both anomaly prediction and detection tasks. Our framework breaks through the limitations of traditional anomaly detection, enhancing the capability to predict future anomalies.

## REPRODUCIBILITY STATEMENT

In this work, we have made every effort to ensure reproducibility. Reproducing the results would be straightforward and require minimal additional effort, ensuring high reproducibility. In the anonymous repository link we provide, you will find the code and evaluation datasets, which are well-documented and easily accessible. Within the code files, we provide instructions to facilitate running and reproducing our experimental results. In Methodology 3, from input to output, we provide a detailed description of our method, including model structure, model training, and anomaly criterion. In Appendix A.3, we summarize all the default hyper-parameters, including noise ratio, learning rate, etc.

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

# A    TRAINING DETAILS

## A.1    DATASETS

In our study, we represent the number of samples in the training set, validation set, and test set with the labels "Train", "Valid", and "Test", respectively. The 'Dim' column indicates the dimension size of the data for each dataset. Additionally, the 'AR' (anomaly ratio) column denotes the proportion of anomalies within the entire dataset.

Table 5: Details of original datasets.

| Datasets | Domain | Train | Valid | Test | Dim | AR (%) |
|---|---|---|---|---|---|---|
| MSL | Spacecraft | 46,653 | 11,664 | 73,729 | 55 | 10.5 |
| SMAP | Spacecraft | 108,146 | 27,037 | 427,617 | 25 | 12.8 |
| SWaT | Water treatment | 396,000 | 99,000 | 449,919 | 51 | 5.78 |
| NIPS-TS-SWAN | Space Weather | 48,000 | 12,000 | 60,000 | 38 | 23.8 |
| NIPS-TS-GECCO | Water treatment | 55,408 | 13,852 | 69,261 | 9 | 1.25 |
| SMD | Server Machine | 566,724 | 141,681 | 708,420 | 38 | 4.2 |
| PSM | Server Machine | 105,984 | 26,497 | 87,841 | 25 | 27.8 |

## A.2    BASELINES

- CAE-Ensemble (Campos et al., 2022) proposed CNN-based autoencoder and diversity-driven ensemble learning method to improve the accuracy and efficiency of anomaly detection.

- GANomaly (Du et al., 2021) adopts the network structure of Encoder1-Decoder-Encoder2, and introduces the idea of adversary-training to provide unsupervised learning scheme for anomaly detection.

- OmniAnomaly (Su et al., 2019) further extends the LSTM-VAE model to capture temporal dependencies in the context of random variates, using reconstruction probabilities for detection.

- Anomaly Transformer(Xu et al., 2021) focuses on the relationship between adjacent points and designs a minimax strategy to amplify normal-anomaly resoluteness of association differences.

- $D^3R$(Wang et al., 2024) performs unsupervised anomaly detection on unstable data by decomposing it into stable and trend components, directly reconstructing the data after it has been corrupted by noise.

- DCdetector (Yang et al., 2023) proposes a contrastive learning-based dual-branch attention structure without considering reconstruction errors. This structure is designed to learn a permutation invariant representation that enlarges the representation differences between normal points and anomalies.

- IForest (Liu et al., 2008) directly describes the degree of distance between points and regions to find abnormal points.

- PAD (Jhin et al., 2023) presents a neural controlled differential equation-based neural network to solve both anomaly detection and Precursor-of-Anomaly (PoA) detection tasks.

- DAGMM (Zong et al., 2018) is a deep automatic coding Gaussian mixture model comprises a compression network and an estimation network. Each network measures information necessary for the anomaly detection.

- Deep SVDD (Ruff et al., 2018) finds an optimal hypersphere in a feature space trained using a neural network. Normal data points are concentrated as much as possible within the hypersphere. Points that are far from the center of the sphere are considered anomalies.

## A.3 IMPLEMENTATION

In our experiments, we set the blocks of the Transformer to 3. We select different sliding window size options for different datasets. For the contrastive branch, we set the noise ratio to 50% for generating samples across different scale sequences. The prediction window size is uniformly set to 4 following previous works (Jhin et al., 2023; Yin et al., 2022). Early stopping with the patience of 3 epochs is employed using the validation loss. We use the Adam optimizer with a learning rate of 1e-5. All experiments in this work are conducted using Python 3.9 and PyTorch 1.13 (Paszke et al., 2019), and executed on CUDA 12.0, NVIDIA Tesla-A800 GPU hardware.

## B COMPARISON WITH MAINSTREAM METHODS

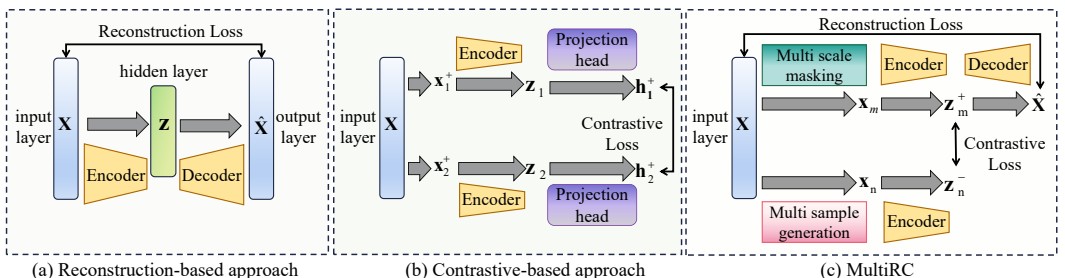

(a) Reconstruction-based approach     (b) Contrastive-based approach     (c) MultiRC

Figure 6: Architecture comparison of three approaches. $\mathbf{h}_1^+$, $\mathbf{h}_2^+$, $\mathbf{z}_\mathbf{m}^+$ are positive samples, $\mathbf{z}_n^-$ is negative samples.

In time series anomaly detection, reconstruction-based methods(Figure 6) train models to accurately reconstruct normal samples, while those that cannot be accurately reconstructed are considered anomalies. This approach is popular because of its ability to combine with multiple machine learning or deep learning models to effectively capture complex dependencies in time series and provide intuitive explanations for anomaly detection. However, they usually do not pay attention to the trend of changes in the data over a period of time, resulting in insensitive recognition of abnormal precursor signals. In addition, it is often difficult to strike a balance between the complexity of the model and the ability to reconstruct it, too simple models can not capture the time dependence, and too complex models tend to lose the ability to distinguish abnormal data. This makes it more difficult to learn a model that is flexible enough to capture time series fluctuations while maintaining good reconstruction performance.

In recent years, contrastive representative learning has emerged in the field of anomaly detection, which can detect anomalies without the need for high-quality reconstruction models. The key idea is that normal data has strong correlation with other data, while abnormal data has weak correlation with each other. However, due to the lack of negative sample labels, the features learned by such methods will fall into a single pattern, resulting in the inability to recognize temporal fluctuations. MultiRC establishes a dual-branch with joint reconstructive and contrastive learning upon a multi-scale structure, adaptively recognizing varying reaction times for different variables through the adaptive dominant period mask. Additionally, hard negative samples are constructed to prevent model degradation. The rich representation learned by the model is conducive to capturing the fluctuations of the time series and the degree of fluctuations, ensuring the effect of the prediction.

## C SCALES ANALYSIS

Multi-scale approaches have a unique impact on anomaly prediction problems, particularly concerning reaction times of varying lengths. The prediction and detection performances at different scales are shown in Figure 7 and Figure 8, respectively.

First, a single scale is suitable for specific ranges of reaction times but has limited performance. Second, the dependency on different scales varies across datasets. Using the PSM dataset as an example, the performance decline is greater when using the 2,8 or 4,8 scale combinations compared

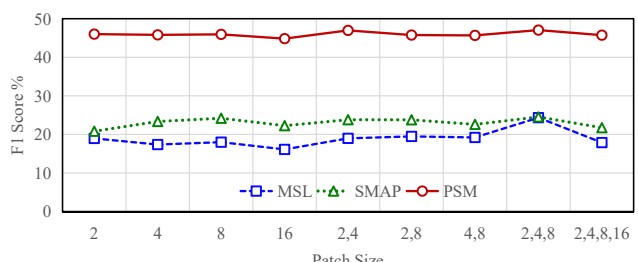

Figure 7: Anomaly prediction results at different scales (F1).

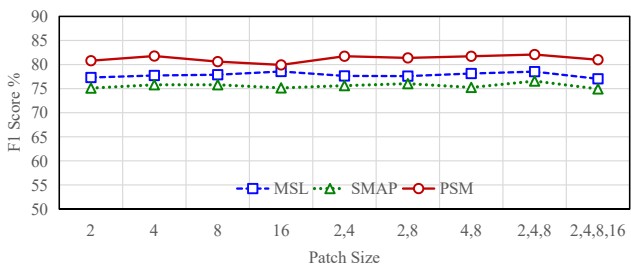

Figure 8: Anomaly detection results at different scales (Aff-F1).

to the 2,4 scale combination. This indicates that shorter time scales are more suitable for anomaly prediction in this dataset. Moreover, more scales are not necessarily better. When the gap between the smallest and largest scales becomes too large, it increases the difficulty of contrastive learning.

## D  SAMPLE GENERATION

Sample generation methods include amplitude magnification or reduction by some factor (scale), reducing resolution (compress), mirroring on the mean value (horizontal axis) (hmirror), temporal displacement by a fixed constant (shift), adding Gaussian white noise (noise), and reversal on the time axis (vmirror). To better reflect real-world conditions, the noise magnitude is randomly selected. The abnormal prediction results under different generation methods are shown in the Figure 9.

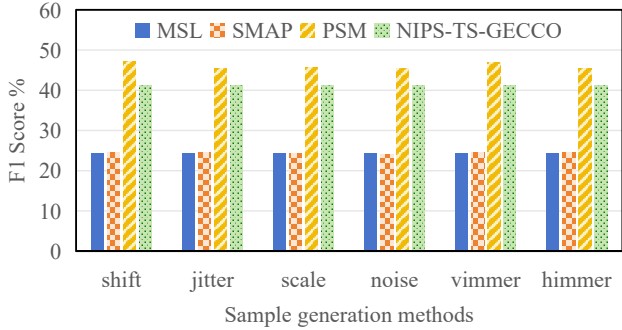

Figure 9: F1 scores for six sample generation methods.

Taking scale as an example, by a given probability, each original data point is randomly magnified or reduced, causing positive and negative samples to vary in scale at different points. The data after noise addition is:

$$\mathbf{x}_{n,p} = \mathbf{x}_p \cdot (\sigma_p \cdot s_p + 1) \tag{13}$$

where $s_p$ represents the random factor generated for scale $p$, and $\sigma_p$ is the coefficient controlling the noise intensity. The random factor is sampled from a standard normal distribution to ensure it is

Table 6: Anomaly prediction results for the NIPS-TS datasets. All results are in %.

| Method | Dataset | ACC | P | R | F1 | AUC |
|---|---|---|---|---|---|---|
| | A.T. | 58.19 | 52.90 | 39.77 | 45.50 | 52.98 |
| | DCdetector | **74.83** | 21.04 | 10.94 | 14.40 | 50.58 |
| NIPS-TS-SWAN | CAE-Ensemble | 58.57 | 56.33 | 44.25 | 49.56 | 64.81 |
| | D$^3$R | 58.38 | 54.96 | 44.81 | 49.37 | 63.87 |
| | MultiRC | 61.75 | **71.28** | **55.22** | **62.23** | **66.82** |
| | A.T. | 89.41 | 43.62 | 27.92 | 34.05 | 50.39 |
| | DCdetector | 97.21 | 1.76 | 3.46 | 2.33 | 50.79 |
| NIPS-TS-GECCO | CAE-Ensemble | 97.71 | 51.67 | 29.92 | 37.90 | 52.94 |
| | D$^3$R | 98.01 | 51.24 | **30.91** | 38.56 | 53.09 |
| | MultiRC | **99.10** | **82.89** | 27.39 | **41.17** | **53.35** |

Table 7: Experimental results for the anomaly detection on NIPS-TS datasets, are presented in percentages.

| Method | Dataset | ACC | Aff-P | Aff-R | Aff-F1 | AUC |
|---|---|---|---|---|---|---|
| | A.T. | 63.89 | 48.12 | 25.89 | 33.67 | 44.74 |
| | DCdetector | 66.54 | 44.47 | 8.61 | 14.42 | 43.46 |
| NIPS-TS-SWAN | D$^3$R | **68.83** | **68.73** | 31.49 | 43.19 | 37.31 |
| | MultiRC | 67.84 | 63.28 | **41.74** | **50.30** | **59.48** |
| | A.T. | 95.82 | 50.39 | 88.68 | 64.27 | 51.60 |
| | DCdetector | 97.50 | 52.07 | 90.79 | 66.18 | 45.38 |
| NIPS-TS-GECCO | D$^3$R | **98.90** | **77.83** | 79.56 | 78.68 | 80.72 |
| | MultiRC | 97.67 | 71.68 | **97.71** | **82.70** | **93.11** |

suitable for all patches at scale $p$:

$$s_p \sim \mathcal{N}(0, 1) \tag{14}$$

Between different scales, $s_p$ is sampled independently to ensure that each scale uses a distinct random factor. This design strategy enables the model to focus on common features at the same scale.

## E  VISUAL ANALYSIS

We visualize the anomaly prediction of MultiRC on the PSM dataset in Figure 10. The model raises warnings by identifying anomaly precursors (pink regions) before the actual anomalies (red regions) occur. This shows that MultiRC is effective in predicting whether future anomalies will occur.

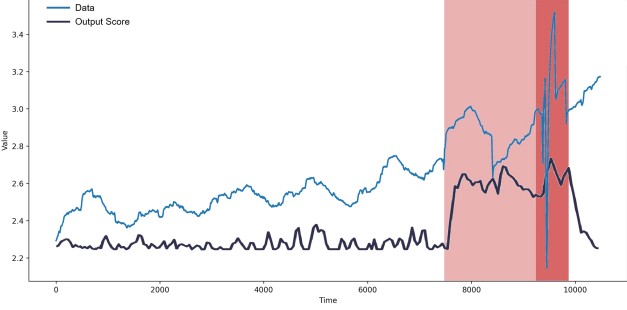

Figure 10: Visualization of the anomaly prediction on the PSM dataset. MultiRC can output larger scores to identify the anomaly precursors that predict the future time series are more likely to be abnormal.

## F  ADDITIONAL PREDICTION RESULTS

We further evaluate MultiRC performance on the NIPS-TS-SWAN and NIPS-TS-GECCO datasets (Table 6), which contain a broader variety of anomaly types. Even in these more complex anomaly

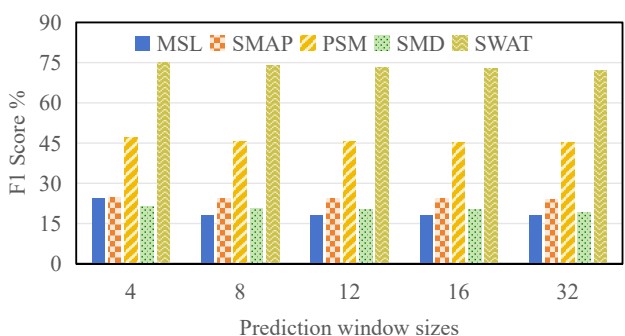

Figure 11: F1 score results for different prediction window sizes.

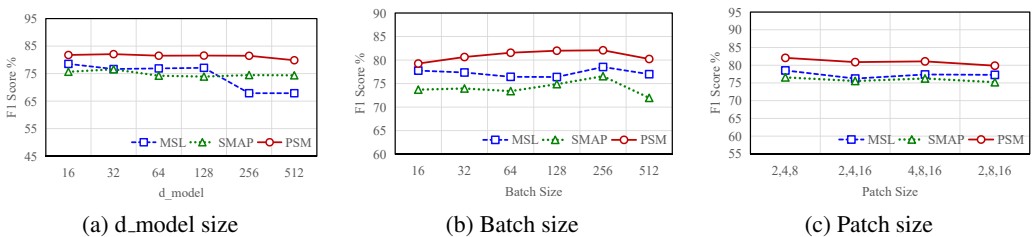

(a) d_model size                      (b) Batch size                      (c) Patch size

Figure 12: For anomaly detection task, parameter sensitivity studies of main hyper-parameters.

environments, MultiRC consistently exhibits superior performance on most metrics when compared to well-performing baseline methods.

To analyze the impact of the prediction distance on the results, we conduct experiments with different prediction window sizes. The results are shown in Figure 11. As the prediction distance increases, the difficulty of prediction gradually rises, which directly leads to a decline in model performance. This phenomenon is in line with expectations, when the prediction window is smaller, meaning the prediction distance is shorter, the fluctuations in reaction time are more closely related to future sub-sequences. However, as the prediction window expands, the model needs to consider data changes and potential dynamic patterns over a longer period, requiring the model to have stronger generalization capabilities and the ability to capture long-term dependencies.

# G   ADDITIONAL DETECTION RESULTS

Table 7 presents a performance comparison between MultiRC and other well-performing baseline methods on the NIPS-TS-SWAN and NIPS-TS-GECCO datasets. Despite the two datasets have the highest (23.68% in NIPS-TS-SWAN) and lowest (1.25% in NIPS-TS-GECCO) anomaly ratio, our model consistently demonstrates good performance across these challenging conditions. On the NIPS-TS-SWAN dataset, our model surpasses the best baseline model by 7.11% in the key metric Aff-F1 (from 43.19 to 50.30). On the NIPS-TS-GECCO dataset, it shows a more significant improvement of 4.02% (from 78.68 to 82.70).

We also study the parameter sensitivity of MultiRC in anomaly detection tasks. Figure 12a shows the performance at different latent space dimensions. Figure 12b also displays the model performance under different batch sizes. The model works well with larger batch sizes. In this study, the design of multiple patch sizes is a crucial element. For our primary evaluation, the patch sizes are typically set to combinations of 2, 4, 8. The results displayed in Figure 12c demonstrate that MultiRC exhibits stable performance across various patch size combinations.

