# OpenReview forum: "MultiRC: Joint Learning for Time Series Anomaly Prediction and Detection with Multi-scale Reconstructive Contrast"
_ICLR.cc/2025/Conference — ICLR 2025 Conference Withdrawn Submission_

### Official Review · Reviewer_Xn2f · 2024-11-01

**Soundness:** 2
**Presentation:** 1
**Contribution:** 2
**Rating:** 3
**Confidence:** 5

**Summary:**

This manuscript focuses on anomaly prediction, which aims to predict whether an anomaly will occur in the future by capturing the fluctuations at the current time. The proposed model is based on contrastive learning and reconstruction loss at multiple resolutions. In the proposed contrastive framework, patches' representations from different resolutions are positive to each other. Negative samples are generated by different noise pollution strategies. Those generated samples are utilized to represent fluctuations for anomaly prediction during training. Finally, anomaly scores are based on the summation of reconstruction loss and the contrastive distances.

**Strengths:**

N/A

**Weaknesses:**

My main concern is the anomaly prediction task mentioned in this paper.

In my opinion, this problem is fundamentally consistent with the anomaly detection task itself.

For example, the yellow area in Figure 1 represents the early anomaly part. Therefore, when the anomaly labels are accurate (indicating the beginning of anomalies regardless of severity), the anomaly prediction task assumed in this paper should essentially belong to the standard anomaly detection task. In other words, the anomaly labels in Figure 1 are inaccurate. It should include the yellow area.
Therefore, I do not think the anomaly prediction problem differs from anomaly detection tasks.

I know that several multivariate anomaly detection datasets used in this paper have inaccurate anomaly labels. The widely used UCR anomaly dataset contains many time series with accurately labeled anomalies. I suggest that the authors use this dataset for further research. Moreover, the paper lacks specific examples of dataset analysis. It would be better to provide more visualized results to support the research.

The paper's methodology is limited in terms of innovation. There have been research works on contrastive representation learning for anomaly detection. The investigation into related works is insufficient. Also, the motivation for using representations from different resolutions as positive samples is unclear.

From the source code, window sizes seem to be too short (64), much less than the actual abnormal window size (>1000).

The writing quality of this version of the paper is unsatisfactory due to many unexplained details. In particular, mathematical formulas and algorithm descriptions lack explanation or contain errors, resulting in some details of the model being unclear.

For examples:
- The subscript range in x_{t-h-1:t-h-1} at line 216 is incorrect.
- It's unclear how patch x_p at line 244 maps multiple scales to different timesteps due to varying patch numbers at different granularities.
-  Formulas 6 lack formal definitions for c and i, with no relevant explanations in the context or formulas.
- It's unclear how x_{n,p} at line 293 is generated, and the definitions of n and p are ambiguous.
- The settings for lambda_{Con} and lambda_{Rec} in (11) are not clearly explained.
- The variables h and f at line 417 are not utilized in the model section, raising confusion. The specific differences between the anomaly prediction process during training/testing and anomaly detection tasks lack explanation.

Since the instance normalization technique is utilized in the proposed framework when some abnormal ts are segmented and shifted from the original range, will this normalization operation produce false negative results?

Computational costs: As you use channel independence, what are the computational costs? Will dealing with multivariate time series be much slower?

**Questions:**

Please see Weakness.

---

### Official Review · Reviewer_uBdV · 2024-11-01

**Soundness:** 4
**Presentation:** 3
**Contribution:** 3
**Rating:** 6
**Confidence:** 4

**Summary:**

This paper introduces MultiRC, a novel approach for joint time-series anomaly prediction and detection. MultiRC combines reconstructive and contrastive learning within a multi-scale framework that adapts to varying reaction times for different time-series variables using an adaptive dominant period mask. The model includes a dual-branch architecture, leveraging asymmetric encoder-decoder networks to fuse cross-scale information. To enhance training momentum and prevent model degradation, MultiRC generates controlled negative samples, unlike traditional contrastive methods that focus solely on positive samples. Evaluated on seven benchmark datasets, MultiRC achieves state-of-the-art performance in both anomaly prediction and detection tasks.

**Strengths:**

1. The paper proposes a novel approach combining anomaly prediction and detection within a joint learning framework, allowing for forecasting anomalies with variable reaction times. Such a perspective is very interesting and would provide insights for the community.
2. For the reconstruction part, the use of a multi-scale structure with an adaptive dominant period mask is innovative; for the contrastive learning part, the generative strategy for creating negative samples also seems novel.
3. The authors provide a well-structured codebase that is easy to navigate and highly accessible for reproduction and further experimentation. The experimental design is very thorough, with sensitivity, scale and ablation study.

**Weaknesses:**

- The paper does not explicitly address the computational time and resource demands introduced by additional components, such as the negative sample generation strategy. Including an analysis or discussion of these overheads would provide valuable insights.
-  The dual-branch, multi-scale structure with asymmetric encoder-decoder networks and generative strategies may be overly complex, which could lead to difficulties in model training, interpretability, and implementation. It would be helpful if the author touches upon this aspect.

**Questions:**

- Did you consider how the complexity of the model architecture affects its interpretability in real-world implementation?
- Given the complexity of the dual-branch, multi-scale structure, did you encounter any specific challenges during model training? How feasible is it to train this model on larger datasets or in resource-constrained environments?
- Can you provide a discussion of the computational time and resource demands associated with the added components?

---

### Official Review · Reviewer_A2Wy · 2024-11-02

**Soundness:** 2
**Presentation:** 2
**Contribution:** 2
**Rating:** 6
**Confidence:** 5

**Summary:**

The paper introduces MultiRC, a novel framework designed to address the challenges in both time series anomaly detection and prediction. MultiRC innovatively integrates reconstructive and contrastive learning approaches within a multi-scale structure to adapt to varying reaction times in time series data. The framework employs an adaptive dominant period mask to recognize different scales of reaction times and uses controlled generative strategies to create negative samples, which enhances the model's ability to learn meaningful representations and prevents degradation.

**Strengths:**

The writing is clear and easy to understand.
The pictures and tables in the paper are beautiful and clear.

**Weaknesses:**

1. Lack of Clarity in Anomaly Prediction Module: The paper does not distinctly outline the modules dedicated to anomaly prediction within the proposed framework. This lack of specificity could hinder readers from understanding how the framework directly addresses the anomaly prediction problem.

2. Inaccurate Representation of Prior Work: The description of the Anomaly Transformer seems to overlook its handling of "prior-association" and "series-association," which are crucial aspects of its approach. This misrepresentation could lead to an incomplete comparison with the existing literature. The paper mentions the DCdetector's architecture but does not elaborate on its multi-scale sampling structure and approach. This omission might affect the comprehensiveness of the discussion on related work.

3. Potential Error in Problem Definition: In the anomaly prediction task, the definition of the antecedent range as t-r: r seems to have an error. The starting index of the future look-forward window should be t instead of t+1, as t represents a time point, not a duration.

5. Vagueness in Multi-scale Structure Explanation: The paper states that variables with longer periodic changes are more likely to evolve gradually, but it does not provide clear evidence or examples to support this claim. This lack of clarity could confuse readers and detract from the paper's credibility.

6. Insufficient Detail on Univariate Sub-sequences: The process of obtaining univariate sub-sequences and the rationale behind using similarity calculations for extracting dominant periodic components require further explanation. The current exposition is ambiguous and may impede reader comprehension.

7. Suspicious Performance Data: The PAD method's reported F1 score on the SWaT dataset is lower than both its Precision and Recall values, which is counterintuitive. This discrepancy raises doubts about the accuracy of the experimental data presented in the paper.

**Questions:**

1. Supporting Evidence for Multi-scale Claims: Can the authors provide examples or data to support their claim about the relationship between periodicity and the evolution rate of variables?
2. Detailed Process of Univariate Sub-sequences: Could the authors elaborate on the process of obtaining univariate sub-sequences and the motivation behind using similarity calculations for extracting dominant periodic components?
3.Verification of Experimental Data: Can the authors explain the discrepancy in the PAD method's F1 score on the SWaT dataset? Is there a possibility of an error in the reported data, and if so, could they provide the correct values?
4.Could the authors provide more detailed information on the specific operations for constructing anomaly samples, such as whether different construction operations were performed for different datasets: what kinds of construction methods were used, and how intense the construction was? Can this information be supplemented in the hyper parameters provided in the paper?

---

### Official Review · Reviewer_krP3 · 2024-11-02

**Soundness:** 3
**Presentation:** 3
**Contribution:** 2
**Rating:** 5
**Confidence:** 4

**Summary:**

This paper introduces MultiRC, a framework for unsupervised time series anomaly prediction and detection that leverages a multi-scale structure and sample generation techniques. Experiments were conducted on seven datasets to evaluate the effectiveness of MultiRC.

**Strengths:**

1. The problem (anomaly prediction) studied in this work is interesting.

2. The experiments were conducted on seven datasets.

3. A joint learning framework of anomaly prediction and detection is proposed.

**Weaknesses:**

1.  Some important related work are missing, especially the work on anomaly precursor detection or early anomaly detection.

 - Unsupervised Time Series Anomaly Prediction with Importance-based Generative Contrastive Learning, Kai Zhao et al.
 - dCNN/dCAM: anomaly precursors discovery in multivariate time series with deep convolutional neural networks, Paul Boniol et al.
 - Deep Multi-Instance Contrastive Learning with Dual Attention for Anomaly Precursor Detection, Dongkuan Xu et al.
 - Early anomaly detection in time series: a hierarchical approach for predicting critical health episodes, Vitor Cerqueira et al.

2. One Url provided by the authors could potentially be used to find authors’ identity.

From the readme of the code Url the authors provided (https://anonymous.4open.science/r/MultiRC-CCE6/README.md): all benchmarks are provided in their Google Cloud (https://drive.google.com/drive/folders/1RaIJQ8esoWuhyphhmMaH-VCDh-WIluRR). But from the Google Cloud link, both the owner/author's photo and username appears there, which might violate the anonymous Url rule.

3. Writing or presentation issues:
- Some full names are not provided: e.g., Aff-F1, Aff-P, Aff-R.
- Figure 2 is too complex to understand and it is not self-contained.
- Table 2: Best results are not highlighted.

4. Experimental issues:
 - The results on all metrics including F1, Precision, recall, and AUC should be shown for all datasets.
 - More showcases with different types of precursors are needed to demonstrate the effectiveness of the proposed method.

5. Formal definition of precursor is needed.
 - The differences between the precursor and the real anomaly are not clear based on current definition.

**Questions:**

1. F1, Precision, recall, and AUC results for all the datasets

2. The differences between the precursor and the real anomaly? And the different types of precursors?

---

### Official Review · Reviewer_fum5 · 2024-11-08

**Soundness:** 2
**Presentation:** 2
**Contribution:** 2
**Rating:** 3
**Confidence:** 3

**Summary:**

This paper proposed a multiscale approach called MultiRC for anomaly prediction and detection. MultiRC integrates reconstructive and contrastive learning and utilizes multi-scale structure and adaptive dominant period mask to deal with the diverse reaction time. It is evaluated on 7 benchmark datasets for both anomaly prediction and detection.

**Strengths:**

1. Good numerical results on anomaly detection tasks.
2. Good writing, making the proposed method understandable from a reproducibility perspective.

**Weaknesses:**

1. My biggest concern about this work is the vague definition of “anomaly”. If we impose a more s, then more points/subsequence including the so-called reaction time can be classified as anomalies. The setting of the reaction time and the “real” anomaly makes it hard to understand the setting of the problem to be solved.
2. I would recommend reformulating the problem as a specific pattern prediction problem.

**Questions:**

1. What’s the definition of anomaly in your work? Can you give some formal definition? Is it possible to define it a temporal pattern problem?

---

### Note · Authors · 2024-11-26

I have read and agree with the venue's withdrawal policy on behalf of myself and my co-authors.